# Oxidative Stress in Obstructive Sleep Apnea Syndrome: Putative Pathways to Hearing System Impairment

**DOI:** 10.3390/antiox12071430

**Published:** 2023-07-15

**Authors:** Pierluigi Mastino, Davide Rosati, Giulia de Soccio, Martina Romeo, Daniele Pentangelo, Stefano Venarubea, Marco Fiore, Piero Giuseppe Meliante, Carla Petrella, Christian Barbato, Antonio Minni

**Affiliations:** 1Division of Otolaryngology-Head and Neck Surgery, Ospedale San Camillo de Lellis, ASL Rieti-Sapienza University, Viale Kennedy, 02100 Rieti, Italy; 2Division of Clinical Pathology, Director of analysis Laboratory of De Lellis Hospital, Viale Kennedy, 02100 Rieti, Italy; 3Institute of Biochemistry and Cell Biology (IBBC), National Research Council (CNR), Department of Sense Organs DOS, Sapienza University of Rome, Viale del Policlinico 155, 00161 Roma, Italy; 4Department of Sense Organs DOS, Sapienza University of Rome, Viale del Policlinico 155, 00161 Roma, Italy; 5Clinical Pathology Physician, Director of Analysis Laboratory of De Lellis Hospital, Division of Otolaryngology-Head and Neck Surgery, Ospedale San Camillo de Lellis, ASL Rieti-Sapienza University, Viale Kennedy, 02100 Rieti, Italy

**Keywords:** OSAS, oxidative stress, hearing, auditory, biomarker, otorhinolaryngology

## Abstract

Introduction: OSAS is a disease that affects 2% of men and 4% of women of middle age. It is a major health public problem because untreated OSAS could lead to cardiovascular, metabolic, and cerebrovascular complications. The more accepted theory relates to oxidative stress due to intermittent hypoxia, which leads, after an intense inflammatory response through multiple pathways, to endothelial damage. The objective of this study is to demonstrate a correlation between OSAS and hearing loss, the effect of the CPAP on hearing function, and if oxidative stress is also involved in the damaging of the hearing system. Methods: A review of the literature has been executed. Eight articles have been found, where seven were about the correlation between OSAS and the hearing system, and only one was about the CPAP effects. It is noted that two of the eight articles explored the theory of oxidative stress due to intermittent hypoxia. Results: All studies showed a significant correlation between OSAS and hearing function (*p* < 0.05).Conclusions: Untreated OSAS affects the hearing system at multiple levels. Oxidative stress due to intermittent hypoxia is the main pathogenetic mechanism of damage. CPAP has no effects (positive or negative) on hearing function. More studies are needed, with the evaluation of extended high frequencies, the execution of vocal audiometry in noisy environments, and the evaluation of potential biomarkers due to oxidative stress.

## 1. Introduction

Obstructive sleep apnea syndrome (OSAS) is a respiratory disease relating to sleep. It is characterized by five or more respiratory events (apnea, hypopnea, or RERA), with a specific symptomatology. Every episode of apnea or hypopnea persists for at least 10 s, is associated with a loss of blood oxygenation of 3–4%, and ends with a short and unconscious awakening from sleep. A population of 2% of women and 4% of men older than 50 years old is often affected by this pathology [1]. Car accidents and cardiovascular morbidity and mortality are the main consequences of untreated OSAS. The pathology is exacerbated by alcohol consumption, sedative use, and weight gain. Obesity is one of the main risk factors for OSAS [2]. In adults, multiple craniofacial anomalies are associated with OSAS, including the increased distance of the hyoid bone from the mandibular plane, decreased mandibular and maxillary projection, a downward and posterior rotation of mandibular and maxillary growth, increased vertical facial length, increased vertical length of the posterior airway, and increased cervical angulation [3]. OSAS is classified as an apnea hypopnea index (AHI) of 5–15 indicating mild OSAS, 15–30 moderate OSAS, and more than 30 severe OSAS [4]. The etiology and the mechanism of the collapse of the upper airway are multifactorial, even though the main feature is the interaction between the propensity of the upper airways to collapse and the distension of the dilator’s pharyngeal muscles. Furthermore, the reflex pathway from the central nervous system to the pharynx can fail in maintaining the patency of the upper airway [5]. The three main obstruction areas are nose, palate, and hypopharynx. The most common OSAS symptoms are snoring, restless sleep, and daytime sleepiness. Sleepiness and fatigue might be exacerbated by multiple other medical conditions that should be considered in a patient suspected of OSAS. With the increasing prevalence of OSAS in patients affected by cardiovascular diseases, cerebrovascular accidents, and diabetes mellitus, these populations should be carefully evaluated for signs and symptoms of OSAS [6]. Physical examination helps to reach the diagnosis, and it includes BMI calculation, blood pressure, neck circumference, body habitus, mandible and maxilla position, facial anomalies, nose and paranasal sinuses evaluation, tongue position and size, elongation of palate and uvula, tonsil size, Mallampati score, dentition, Hyoid, and mandible position, including retrognathia. Nasal and laryngeal endoscopy is important for the evaluation of the upper airways [7]. The Muller maneuver is performed with an awake patient, which produces a negative pressure inhaling against a closed glottis, with a closed mouth and nose, to induce a collapse of the upper airways. To evaluate better the sites of obstruction in OSAS patients, Drug-Induced Sleep Endoscopy (DISE) is the gold standard to guide the physician to the best therapeutic option [8]. DISE is performed with a fiberoptic nasopharyngoscopy to evaluate the region of the respiratory collapse, during a drug-induced sleep [9]. To standardize the findings during DISE, VOTE classification (Velum, Oropharynx, Tongue base, Epiglottis) is a more reliable tool to predict surgery outcomes [10]. Nocturnal PSG is the gold standard for the diagnosis. OSAS treatment should be approached gradually. Weight loss should be recommended to all OSAS patients that are overweight. CPAP is the gold standard for the treatment of moderate and severe OSAS. [11]. Multiple studies have demonstrated its efficacy in reducing the AHI index, improving sleeping and quality of life in general, and reducing the risk of cardiovascular morbidity and car accidents. Oral applications are recommended in patients with mild and moderate OSAS and, in some cases, in patients who cannot tolerate the CPAP [12,13]. Regarding surgery, the site of obstruction should be identified in every patient to address the type and the extent of the surgical procedure [14]. Septoplasty, turbinate reduction, nasal valve surgery, and sinus surgery are procedures with the purpose of treating the nasal obstruction associated with OSAS [15]. However, this type of surgery alone does not significantly improve the disease. Improving nasal patency is helpful in the recovery of physiological breathing and tolerating CPAP [16]. Uvulopalatopharyngoplasty (UPPP) associated with bilateral tonsillectomy, by resection of redundant mucosa and pharyngeal tissue, has been developed to treat palatal obstruction [17]. This procedure should be limited to patients with oropharyngeal obstruction. Lateral pharyngoplasty was described for the first time in 2003 as an alternative to UPPP [18]. This study showed a significant decrease in the AHI index associated with an improvement in the quality of sleep and daytime sleepiness. Barbed pharyngoplasty demonstrates effectiveness and safeness with a significant improvement in the patient’s symptoms and AHI [19,20]. Partial midline glossectomy, lingual plasty, and radiofrequency ablation of the tongue base have been developed to treat retro-lingual collapse. Lingual tonsillectomy could be useful in patients affected by lingual tonsillar hypertrophy. In selected cases, lingual tonsillectomy, reduction in the aryepiglottic folds, and partial epiglottectomy could be performed [21]. However, because of the significant incidence of postoperative edema, these procedures are often executed in combination with a tracheotomy to protect the upper airway. Other surgical treatments are genioglossal advancement and hyoid advancement, which have the aim to enlarge the retro lingual space. Transoral robotic surgery (TORS) has been shown to be effective and safe [22]. Untreated OSAS could lead to cardiovascular, metabolic, and cerebrovascular comorbidity. The pathophysiology of these complications is not entirely elucidated but seems to involve multiple pathways, one of which is endothelial damage due to oxidative stress. This is defined as an imbalance between the pro-oxidant and antioxidant system, which leads to the excessive formation of reactive oxygen species (ROS). ROS and oxidative stress are tightly associated with hypoxia in OSAS patients [23,24]. Oxidative stress is one of the most important features in the development of cardiovascular comorbidity. Oxidative stress and intermittent hypoxia could lead to multiorgan impairment. One of these organs susceptible to oxidative stress is the auditory system. The cochlear hair cells are most vulnerable to oxidative stress, particularly those located at the base of the cochlea itself, leading to sensorineural hearing loss (SNHL), especially for high frequencies, in response to multiple causes, such as ototoxic agents, noise exposure, and aging. Antioxidant therapy is promising, suggesting the need to discover new biomarkers for an early diagnostic framework of OSAS patients [25]. Antioxidant therapy has also proven to be effective for acquired disorders that induce SNHL [26]. The objective of this review is to explore a correlation between OSAS and the auditory system. More specifically, our aim is to underline that untreated OSAS could negatively affect the auditory function as compared to non-OSAS patients. Hearing impairment risk factors, such as age, ototoxic drugs, noise, diabetes, altered lipid metabolism, smoking, and coronary heart disease, can be evaluated in OSAS patients with the aim being to quantify more specifically state the possible risk factors of the auditory function in OSAS patients as compared to non-OSAS patients. Several studies have focused on risk factors for OSAS or hearing loss, but no studies have been conducted for both combined and associated diseases. We suggest that OSAS itself could be a risk factor for hearing loss. Antioxidant therapy could be effective in OSAS and in SNHL, but to date there is no evidence of the effectiveness of antioxidant therapy in OSAS patients with SNHL. Moreover, we discuss if the CPAP could improve patients’ hearing, or if its prolonged application could worsen the auditory system. It will also explore the pathophysiology of auditory damage, showing several pathogenic theories. Finally, it will be explored if oxidative damage, as in the other comorbidities, could be part of the impairment of the auditory system in OSAS patients and how antioxidants could be part of the treatment in the future.

## 2. Materials and Methods

A literature review has been conducted. PubMed research with keywords such as “OSAS”, “auditory”, “hearing”, “audiology”, “CPAP”, and “oxidative” has been performed. “OSAS and auditory” retrieved 18 articles, whereas no articles were found under “OSAS and auditory and oxidative”; “OSAS and hearing” retrieved 57 articles, and with “oxidative”, no results were retrieved; “OSAS and audiology” showed 42 results, but only one was found under “oxidative”. Lastly, “CPAP and oxidative” retrieved 206 articles, and 15 articles have been selected. All the published articles with pediatric patients have been excluded because this study is based only on adult patients. Furthermore, all the research that included all of the OSAS-related comorbidities, like cardiovascular and metabolic diseases, or every condition that could affect the auditory system, have not been considered. This is because in order to demonstrate the correlation between OSAS and hearing loss, it is necessary to exclude other causes of hearing damage that are pathology related. Similarly, all the articles with patients affected by ORL diseases, like otitis media, tubaric dysfunctions, and nasal obstruction, which can cause neurosensorial, conductive, or mixed hearing loss, have been excluded. Ultimately, all the studies involving lab animals, have been eliminated from the study because our research is focused only on human subjects. From this literature review, eleven articles were considered, and eight extensively evaluated OSAS effects on the auditory system and how this can be linked to microvascular damage of the tissues, and one focused on the effects of CPAP treatment (Table 1).

## 3. Results

From the literature, a few articles focusing articles underline the effect of OSAS on the auditory system. Quiyang fu et al. [27] analyzed 52 patients (34 males and 18 females, ranging from 23 to 39 years old, mean age of 32.2 ± 4 years), of which 31 were included in the OSAS group (21 males and 10 females, mean age 32.5 ± 4.4 years) and 21 were included in the control group (13 males and 8 females, mean age 31.7 ± 3.5 years). No significant difference in the data was found with pure-tone audiometry DPOAe and click-ABR (*p* > 0.05), both in terms of peak (I-III-V) and interpeak latencies. However, it has been found with speech-ABR that OSA patients had longer latency values than the control group, as regards the transient portion of speech-ABR. The statistical analysis results that were significant differences in the latencies between OSA and the control group for peak V, peak C, and peak O (*p* < 0.05). Another study compared 35 patients (20 males and 15 females, between 39 and 48 years old, mean age 44.4 ± 3.9 years old) affected by moderate and severe OSAS with a control group composed of 30 individuals (17 male and 13 females, between 28 and 51 years old, mean age 43.1 ± 2.4 years old) identified as having low-risk OSAS using the stop-bang questionnaire. With pure-tone audiometry, the hearing threshold of the OSAS group was significantly higher than the control group (*p* < 0.001). Even examining the frequencies separately, the hearing threshold of OSAS patients at 500, 1000, 2000, 4000, and 8000 Hz was significantly higher than the control group [28]. Evaluating the results obtained with the TEOAe, no significant variations between the SNR values and signal amplitude has been found. However, the reproducibility test values in both ears were significantly lower in OSAS patients than in the control group (right ear *p* < 0.001, and left ear *p* < 0.05). Moreover, hearing thresholds and PSG data have been compared. No correlations between AHI, SpO2, TST, TST90, mean desaturation, and pure-tone audiometry data have been found. Spinosi et al. [29] analyzed a sample of 80 patients between 45 and 65 years old, of which 30 took part in the control group (mean age 53.7 years old), and 50 took part in the experimental group (15 simple snoring patients, 35 affected by moderate OSAS, mean age 58.9 years old). Potential dependence/independence between the two groups and the degree of hearing loss has been evaluated. As a result, the Cramer’s measure has been used. This index varies from 0 to 1, and 0 indicates any correlation. In this study, the result was 0,073, implying a clear independence. Martines et al. [30] studied a group of 160 patients (103 males and 57 females, between 38 and 55 years old, mean age 46.19 ± 7.04 years old), of which 60 were healthy patients, 58 were affected by mild OSAS, 18 by moderate OSAS, and 24 by severe OSAS. The severity of the disease was significantly higher in male patients (*p* = 0.01). A total of 54 of these patients were affected by tinnitus, without significant difference in sex; 42 subjects affected by tinnitus were OSAS patients, while on the contrary, 12 were not OSAS patients (*p* = 0.004). A mild difference in the prevalence of hearing loss has been found with pure-tone audiometry (*p* = 0.23). A more significant hearing loss has been found in individuals affected by moderate and severe OSAS (*p* = 0.001). More specifically, all groups were characterized by a mean hearing threshold of 25 dB HL for the frequencies 250, 500, 1000, 2000, and 3000 Hz, and progressive hearing loss, with a peak of 60 dB HL for the frequencies 11.200, 12.500, 14.000 and 16.000 Hz in the case of moderate and severe OSAS. Overall, 45% of simple snoring and 24% of OSAS patients passed the test of TEOAe. Moreover, a significant difference between simple snoring and individuals affected by severe OSAS at the frequencies 3000 and 4000 Hz has been found. The analysis of ROM levels revealed a mean value of 323.29 ± 56.87 CARR U, with a significant difference (*p* = 0.004) between severe OSAS patients and simple snoring. Furthermore, a very higher level of ROM has been observed in individuals affected by tinnitus (*p* = 0.035). Casale et al. [31] selected 39 patients (31 males and 8 females, BMI 31.3 ± 7.6 kg/m^2^, mean age 38 years old), of which 18 (15 males and 3 females, AHI > 30, BMI 32.1 ± 6.5 kg/m^2^, mean age 31 years old) was the severe OSAS group, and 21 (16 males and 5 females, BMI 28.8 ± 5.5 Kg/m^2^, mean age 39 years old), were the control group. The hearing threshold of the OSAS group was significantly higher than the control group (*p* < 0.01) at the pure-tone audiometry. Likewise, the values of the reproducibility test of the TEOAe of OSAS patients were lower than the control group (*p* < 0.01). The brainstem auditory evoked potentials (BAEPs) showed that in OSAS patients, there was a significantly prolonged peak latency in the wave I-III-V than simple snoring (*p* < 0.01). Moreover, analyzing the reproducibility of these waves, a significant difference between OSAS and the control group has been found. However, there was no correlation between AHI, mean SpO2, and auditory data as hearing threshold, TEOAe, and brainstem auditory evoked potential data. Wang et al. [32], in a cross-sectional study to value the effects of sex, hypoxia, and arousal in the brainstem auditory evoked potential of OSAS patients, selected 118 individuals, of which 40 (32 males and 8 females) were affected by moderate OSAS, 44 (36 males and 8 females) were affected by severe OSAS, and 34 (22 males and 12 females) were healthy patients. Compared to the control group, women with moderate and severe OSAS had a longer latency of waves I and V, and men with moderate OSAS also had a longer latency of waves I and V, whereas men with severe OSAS had a longer latency of waves I, III, and V. Furthermore, all the possible correlations between BAEPs data with sex, age, ESS (European Social Survey) questionnaire score, AHI, ODI, MSpO2, MmSpO2, SIT90, and AI in OSAS patients have been analyzed. In both ears of OSAS patients, both males and females, the latency of waves I, III, and V were clearly correlated with AHI, ODI, MSpO2, MmSpO2, SIT90, and AI. No correlation between BAEPs data and age, BMI, and ESS questionnaire score has been identified. Iriz et al. [33] compared 21 patients (12 males and 9 females, mean age 47.1 ± 8.2 years old) affected by OSAS and 10 patients (5 males and 5 females, mean age 43.6 ± 3.8 years old) that took part in the control group. The average hearing threshold of all the participants of this study was 25 dB or higher. The average speech discrimination rate of the OSAS group was 96 ± 3.5 in the right ear and 95 ± 4 in the left ear, whereas in the OSAS group, it was 99.2 ± 1.3 in the right ear and 99.4 ± 1.3 in the left ear. The OSAS group responded correctly with a mean score of 21.4 ± 7.3 to 30 different sounds at the DPT and 16.4 ± 8.9 at the FPT, whereas in the control group, the average correct answers were 25.7 ± 3.4 at the DPT and 22.8 ± 3.5 at the FPT. A decrease in the speech discrimination score in OSAS patients has been found (*p* = 0.01). To study the auditory function, pure-tone audiometry, tympanogram, DPOAe, and click-ABR have been performed. Qiuyang Fu et al. [27] added speech-ABR in their evaluation. The research of Spinosi et al. [29], aimed to evaluate the effect of chronic noise due to snoring, and hearing loss relating to exposure to chronic noise, considers all the patients classified according to age and ELI (Early Loss Index) and calculated according to the national protocols of occupational medicine. ELI was calculated by subtracting a corresponding correction number for presbycusis from the 4000 Hz frequency and by evaluating the loss relating to age and sex. Possible hearing loss was classified on a categorical scale (A-B-C-D-E), with A indicating the best and E the worst auditory performance. Martines et al. [28], testing the effects of hypoxia on the auditory system, decided to administer to their patients, pure-tone audiometry, acufenometry, and TEOAe, adding ROM levels measurement through the d-ROM test. This test is based on the reaction of the hydroperoxides of a biological sample with a transition metal (iron), which catalyzes the formation of free radicals, which then oxidizes an alkylamine forming a colored radical detected by photometry at 505 nm. An amount of 10 mL of blood is mixed with 1 mL of an acidic buffer reagent (R2) to release iron from plasma proteins that will react with peroxides of the blood to form free radicals; then, 10 mL of a chromogen reagent (R1 reagent, alkylamine) is added forming a pink-colored derivative. This colored derivative is photometrically quantified, and the optical density is directly proportional to the concentration of ROMs. Reference values are between 250 and 300 Carratelli Units (CARR-U), which are independent of age and sex. Values between 301 and 320 CARR-U indicate a borderline value; 321-340 CARR-U indicates a low-level oxidative stress; 341-400 CARR-U indicates a middle-level oxidative stress; 401–500 CARR U indicates a high level of oxidative stress; and >500 CARR-U indicates a very high level of oxidative stress. Statistical significance for *p* < 0.05 has been obtained. Lastly, our review considered one article about the correlation between CPAP and its effects on hearing function. Deniz and Ersozlu [34] studied possible changes in the auditory system of OSAS patients under CPAP treatment. They recruited 22 individuals affected by OSAS (44 ears examined), of which those who were under CPAP treatment were labeled Group A, and those who were not using CPAP were labeled Group B. The mean age of group A and group B was 57.33 ± 9.31 and 56.30 ± 11.40 years old, respectively. There was not a significant difference in the age of the two groups (*p* > 0.05). Of these 22 patients, 16 were males, and 6 were females. A decrease in the hearing threshold > 10 dB in six ears of group A and in four ears of group B was detected. No change in hearing was found in 34 ears, and none showed an improvement in hearing ≥ 10 dB. According to the chi-square test for the examination of the relationship between a change in hearing and the two groups, no significant changes in hearing (75% group A and 80% group B) in many patients have been found (*p* > 0.05).

## 4. Discussion

How can OSAS affect the auditory system? There are several hypotheses mainly associated with microvascular impairment and the damaging effects of reactive oxygen species (ROS). Steiner et al. [35] reported that blood plasma viscosity can be increased in OSAS patients, with an impairment of the microcirculation, and in fact, Bernard et al. evidenced that blood higher viscosity can result in a dysfunction of the hearing system [36]. Instead, it was suggested that exposure to continuous noise due to snoring can result in hearing loss [37]. The most accepted theory is that hypoxemia can be a negative factor in OSAS patients, compromising their hearing system. In OSAS disease, hypoxemia causes damage in multiple systems, such as cardiovascular and neurologic diseases. From animal studies, it was reported that intermittent ischemia causes auditory damage due to mitochondrial damage in the hair cells of the internal ear [38]. The internal ear represents an anatomic region extremely susceptible to anoxic/hypoxic damage, due to the formation of ROS and increased oxidative stress, which activates an inflammatory and an immune response leading to both vascular and metabolic complications [39] (Figure 1).

Multiple pieces of evidence suggest that altered mitochondrial function and oxidative stress both in the OSAS and cochlea could play a pivotal role [25,40,41,42,43,44]. These observations were reported individually in OSAS or hearing loss, but investigations aiming to explore the mitochondrial function linking OSAS and hearing system impairment are lacking.

### 4.1. Mitochondrial Dysfunction and ROS

Mitochondria are intracellular organelles that have the function of nutrient metabolization and ATP production, and they are involved in energy metabolism, the generation of free radicals, calcium homeostasis, cell survival, and death [45]. They produce most of the energy of the body in the form of ATP through the Tricarboxylic acid cycle (TCA cycle) and the electron transport chain (ETC). Mitochondria are the major intracellular source of reactive oxygen species (ROS). The electron’s flow through the ETC is an imperfect process, that leads to an incomplete reduction in the oxygen by mitochondria and to the production of ROS [46]. ROS include free radicals, such as superoxide anion (O2•−) and hydroxyl radical (HO•), and non-radicals, such as hydrogen peroxide (H2O2) [47]. It has been shown that the interaction between hydroxyl radicals and DNA causes damage on multiple sites of the DNA itself [48]. ROS imbalance between the production and the level of antioxidant defense in the cell context drives oxidative stress. This oxidative stress causes damage to the mitochondria, leading to an interruption of its functions, such as the production of ATP [49]. There are several defense mechanisms with the aim to protect from oxidative stress which are included enzymatic molecules such as superoxide dismutase, catalase, glutathione reductase, glutathione peroxidase, and non-enzymatic molecules, such as vitamin E and C, glutathione, carotenoids, and flavonoids. Normally, the production of ROS remains in the mitochondria to protect the cells from oxidative damage. However, when the production of ROS overcomes the antioxidant defenses, oxidative stress causes irreversible ROS-mediated cellular damage to DNA, proteins, and lipids [50]. This mechanism damages the respiratory chain and causes mitochondrial dysfunction and leads to multiple pathological conditions like aging, metabolic disorders, and neurodegenerative pathologies. The role of mitochondria in recurrent hypoxia is not only evidenced in neuronal cells, but also in genioglossus and palatine muscles [51]. Mitochondria are the main producers of ROS during reoxygenation [52] (Figure 2). OSAS patients present a reduction in blood mitochondrial DNA levels, and it is a marker of mitochondrial damage [38]. Another source of ROS, during the reoxygenation phase, is the production of superoxide due to enzymes like xanthine oxidase, nitric oxide synthase, and NADPH oxidase [53]. ROS stimulates the expression of adhesion molecules, like L-selectin and integrins, and related endothelial adhesion molecules, like E-selectin, P-selectin, ICAM-1, and VECAM-1, which leads to microvascular damage [54,55]. The imbalance between increased ROS levels and ineffective antioxidant capacity can be quantified by multiple biomarkers, which are proteins generated by the oxidation of nucleic acids, proteins, and lipids [56,57,58,59].

### 4.2. Hypoxia and HIF-α

Our body tries to adjust itself in this condition of hypoxia, by producing numerous molecules useful for cells survival in the condition of a lack of oxygen, like Hypoxia-Induced Factor 1-α (HIF-1α) and Vascular Endothelial Growth Factor (VEGF) [60]. HIF-1α is one of the most important factors in oxygen homeostasis; in fact, its levels, like those of NF-kB, are associated with the severity of the disease (AHI and ODI) [61]. HIF-α is involved in OSAS redox signaling, which leads to multiple systemic and cellular functional changes, like blood pressure, increased release of neurotransmitters, and alteration in sleep and cognitive functions [62]. Sies et al. showed that HIF-α is activated by a different pathway in intermittent hypoxia than in prolonged hypoxia [63]. Intermittent hypoxia seems to cause less stability of HIF-α, leading to the activation of NF-kB, probably due to oxidative stress [64]. NF-kB is probably very important in the pathogenesis of OSAS because it is under complex regulation by multiple regulatory molecules and coordinates the inflammatory response, like the production of adhesion molecules, cytokines, and adipokines [65]. Sterol regulatory element-binding proteins (SREBPs) are a group of transcription factors involved in lipid homeostasis that are redox sensitive [66]. Multiple studies have demonstrated that hyperlipidemia mediated by SREBPs is involved in lipid peroxidation and atherosclerosis induced by intermittent hypoxia [67]. Lipid peroxidation causes impairment in membranes, lipoproteins, and other lipid molecules. Lipid peroxidation causes the production of multiple secondary molecules like some aldehydes [68]. In this regard, malondialdehyde (MDA) is the main product of these aldehydes in the peroxidation mechanism, and it is a marker of oxidative stress in some lung diseases [69]. Pau et al. showed that MDA levels were higher in OSAS patients compared to non-OSAS individuals suggesting MDA as a potential biomarker of OSAS [70]. VE-cadherin cleavage is related to endothelial impairment in OSAS patients, because of increased values of its soluble form in the patient’s blood, leading to increased endothelial permeability and its association with other mechanisms involved with oxidative stress, ROS production, HIF-α, VEGF, and tyrosine kinase pathways [71] (Figure 2). The production of extracellular vesicles by red blood cells is another factor involved in endothelial impairment and relates to decreased eNOS, decreased Endothelin-1 (ET-1), and phosphorylation by the PI3K/AKT pathway [72].

### 4.3. OSAS, miRNAs, and Auditory System

It also seems that some epigenetic alterations are involved in intermittent hypoxia and OSAS, involving the small non-coding-RNA as microRNA (miRNAs) [25]. MiRNAs in this era of medicine are considered ideal biomarkers [73], and it has been found that in OSAS patients, there are decreased levels of miR-199-3p, 107, and 485-5p and increased levels of miR574-5p [74]. In a murine model, it has been evidenced that miR-155 induces oxidation and intensifies the NLRP3 inflammasome pathway induced by intermittent hypoxia, by inhibition of the FOXO3a gene and HK-2 cells [75]. Moreover, miR-155 seems to have a proapoptotic function in some diseases where there is a decreasing level of antiapoptotic molecules, like clusterin, which has increased levels in OSAS patients and correlates with miR-155 [76]. MiR-664a-3p is a potential biomarker of atherosclerosis in OSAS because it has decreased levels in OSAS patients, and it is negatively related to AHI and carotid intima-media maximum thickness [77]. Other micro-RNA, like miR-630 in pediatric OSAS, and miR-30a, miR-34a-5p, and miR-193 in murine studies, take part in endothelial dysfunction in OSAS patients [78,79]. It is to be noted that a deep analysis of altered expression of miRNAs in OSAS and the auditory system could open a new hypothesis to explore the redox unbalancing shared in this connection.

### 4.4. OSAS and Interleukins

Intermittent hypoxia and oxidative stress activate an immune response by our organism, involving proinflammatory molecules, like TNF, CRP, and IL-6 and IL-8, with increased levels of NF-kB and TNF-α [80]. IL-8 and IL-17 are linked to OSAS severity [81]. CRP and TNF- α levels are lower after surgery but still higher than in healthy patients, as shown by Olszewska et al. [82]. Moreover, several peripheral blood cells are involved. Monocytes overexpress Toll-Like Receptor (TLR) in OSAS patients, and the TLR-6 gene is upregulated through methylation of the DNA [83]. Huang et al. [84] showed that site number 1 cytosine-phosphate-guanine (CPG) is hypermethylated in patients affected by severe OSAS. Macrophages also play a key role in this inflammatory process due to oxidative stress, producing numerous molecules and reactive oxygen species, like substances reactive to thiobarbituric acid, 8-OHdG, and asymmetric dimethylarginine [85]. Oxidative stress contributes to sleep behavior in patients with OSAS, and some authors suggest that the intake of antioxidants improves sleep quality [25,86]. However, most of the molecules are not tested in humans, and those tested in humans have not been studied in cohorts large enough to give indications of their use. Vitamin C and N-acetylcysteine showed interesting results regarding the reduction in oxidative stress in OSAS patients [58], evidencing an improvement in sleep parameters. Moreover, it has been demonstrated that leptin can reduce free radicals, oxidative stress, and atherosclerosis in OSAS patients [87].

### 4.5. Oxidative Stress and Auditory System

Regarding the hearing system, the external ciliate cells of the cochlea, at its basal turn, are vulnerable to oxidative stress, probably because there is a minor activity of the antioxidant enzymes correlated with glutathione [88]. It is important to note that according to the tonotopic theory, that the basal turn encodes the high frequencies. Excessive oxidative stress can lead to a dysfunction of the microcirculation, as shown by Patt et al. [89], who examined the endothelial function in OSAS patients not affected by cardiovascular diseases. They evidenced an increase in the peroxynitrite levels in the microvascular wall of the OSAS patients, leading to the excessive production of NO and superoxide in the endothelial environment. This uptake of NO by the superoxide leads to a reduction in the availability of NO and then to changes in the microcirculation that are independent of age, sex, and weight and that are reversible with therapy. Similar mechanisms can be responsible for the progressive damage of the internal ear, where the production of NO by the vascular cells of the cochlea leads to a relaxation of the smooth muscles and pericyte, inhibiting the voltage-gated calcium channels and activating the ATP-sensitive K+ channels in the endothelial and smooth muscles cell of the spiral modular artery [68]. Thus, cochlear damage and hearing loss can be early markers of impairment of the microcirculation in individuals affected by OSAS [90]. CPAP is the gold standard for OSAS treatment, and it is used to overcome rhino- and oropharyngeal collapse in patients with moderate and severe OSAS. The purpose of the CPAP is to allow the patient, during sleep, to breathe and to increase oxygen levels. It is important to know that CPAP is the only confirmed antioxidant therapy and has been shown that it is capable of reversing some of the alterations induced by oxidative stress, like eNOS, nitro-tyrosine, and NF-kB in the endothelium and circulating TNF-α [91]. However, possible side effects can be aerophagy, sinusitis, dryness of the oral and nasal mucosa, congestion, sneezing, and epistaxis. In addition, unexpected and serious side effects have been reported, like pneumocephalus, pulmonary barotrauma, intraocular hypertension, subcutaneous emphysema, and hearing loss due to barotrauma. In their study, Quiyang et al. [27] reported that the transient component of speech-ABR is significantly and positively correlated to the AHI of patients with mild and moderate OSAS, whereas these changes, in the conventional click-ABR, cannot be evidenced. They suggest this test as a potential biomarker in the diagnosis and management of OSAS in the early stages. According to Gozeler and Sengoz [28], the hearing system is affected to various degrees in OSAS patients, detecting, mild neurosensorial hearing loss in OSAS patients with respect to healthy ones. Spinosi et al. [29] showed no correlation between hearing loss and exposure to chronic noise due to snoring, even in patients with mild OSAS. Martines et al. [30] underlined the key role of chronic intermittent hypoxia in the development of hearing dysfunctions and a more marked hearing loss in the high frequencies in patients with severe OSAS. Casale et al. [31] confirmed this theory, asserting that hypoxia could be a risk factor for the hearing system, observing a hearing dysfunction in their patients. Similarly, Iriz et al. [33] showed that repeated hypoxic events can cause multisystemic disorders in OSAS patients, even in the hearing system. Wang et al. [32] evidenced some abnormalities in the brainstem auditory evoked potentials in patients with moderate and severe OSAS, obtaining longer latencies of waves I, III, and V, especially in the right ear of male patients. Ultimately, Deniz and Ersozlu [34] have demonstrated that CPAP does not improve the hearing function of OSAS patients, but it has no negative effects too, due to a possible barotrauma.

## 5. Conclusions and Perspectives

### 5.1. High-Frequency Pure-Tone Audiometry

In our opinion, high-frequency pure-tone audiometry could be a more cost-effective tool for the early diagnosis of hearing loss in OSAS patients. According to the tonotopic theory and to the pathophysiology of the possible auditory damage due to oxidative stress caused by intermittent hypoxia in OSAS patients, extended high frequencies might be impaired first by endothelial damage. High-frequency pure-tone audiometry is a tool used to evaluate hearing threshold in the frequency range of 8–20 kHz and detecting hearing loss before the involvement of medium and low frequencies and the impairment of hearing capacity (Figure 2).

### 5.2. The Role of Antioxidant Therapy

Antioxidants are molecules that inhibit ROS production and regulate oxidative stress. They can be endogenous, produced in vivo, and exogenous, taken from the outside. Endogenous antioxidants are the molecules we discussed early in this section. Exogenous antioxidants comprehend water and lipid-soluble molecules. Water-soluble antioxidants are methionine, vitamin C, carnitine, riboflavin, niacin, folic acid, polyphenols, and catechins. Methionine has the function to reduce cholesterol blood levels and to remove ROS [92]. Riboflavin and niacin eliminate lipid peroxides, with the aid of the GSH [93]. Vitamin C reduces toxicity by removing hydroxyl radicals [94]. Folic acid lowers homocysteine levels [95]. Lipid soluble antioxidants are β-carotene, vitamin E, astaxanthin, and coenzyme q10. Vitamin E gives stability to the biological membranes. Coenzyme q10 lowers the levels of vitamin E radicals after ROS removal [96]. There are some antioxidants that are both water and fat soluble, like Gingko Biloba and alpha lipoic acid. The latter has both antioxidant properties and restores the antioxidant ability of glutathione, vitamin A, vitamin E, and vitamin C [97]. It has been demonstrated that vitamins C and E are potential treatments of SNHL. In fact, it seems that vitamins improve hearing function in patients with sudden hearing loss [98]. It has been found, in an animal study, that vitamin E protects hair cells from the ototoxic damage of the cisplatin [99]. For what concerns OSAS patients, it has been shown that vitamin C and N-acetylcysteine reduce oxidative stress [58]. N-acetylcysteine reduces oxidative stress in OSAS patients by reducing peroxidized lipids and increasing glutathione. Furthermore, an improvement in PSG data has been found [86]. Vitamin C improves endothelial function in OSAS patients [100]. Lastly, it has been found that leptin, in OSAS patients, reduces free radicals, oxidative stress, and atherosclerosis [87]. It is clear that antioxidant therapy could be effective in OSAS and in SNHL due to problems like aging, ototoxic drugs, and noise exposure. However, in our review, no evidence of the effectiveness of antioxidant therapy in patients affected by OSAS and SNHL combined has been found (Figure 2).

### 5.3. Perspectives

OSAS is a disease characterized by chronic snoring, episodes of apnea, and episodes of hypopnea that lead to a reduction in blood oxygen levels and arousal. Sleep fragmentation decreases its quality. CPAP, giving positive pressure in the upper airways, overcomes the obstruction relating to the rhino- and oropharyngeal collapse, healing the symptoms relating to the pathology. Untreated OSAS can lead to cardiovascular, metabolic, and cerebrovascular complications, and one of the main mechanisms that leads to these comorbidities is oxidative stress due to intermittent hypoxia, which leads to an intense inflammatory response via multiple pathways. Untreated OSAS can affect the hearing system, and this is evident in the pure-tone audiometry (deficit for high frequencies), DPOAE, TEOAE, and in the brainstem auditory evoked potentials (prolonged latencies of waves I, III, and V). Such modifications are evident in patients with severe OSAS, being more conspicuous. In individuals affected by mild OSAS, these modifications can be detectable only with speech-ABR, and it was suggested [26] that this test could be a potential biomarker in the diagnosis of the pathology in its early stages. Regarding the correlation between OSAS and the auditory system, and the causes of hearing system dysfunctions, we suggest, as shown by Spinosi et al., that chronic exposure to noise due to snoring is not a risk factor; instead, as confirmed by Martines et al. and Casale et al., intermittent hypoxia due to apnea leads to an endothelial impairment due to oxidative stress that leads to an increase in ROS production. Therefore, ROS production via mitochondrial impairment activates a vicious circle, as discussed above. Importantly, even though CPAP treatment does not improve hearing function, its prolonged use does not cause side effects to the hearing system. The limitations of these studies include the fact that they did not report very high frequencies. Iriz et al. did not perform a PSG on the patients involved in the study, and on the control group not presenting OSAS-related symptoms. We propose that for a better understanding of the phenomenon, it is desirable to enlarge the studies. In this study, it has been underlined that OSAS negatively affects the auditory system and that there is a correlation between the severity of the disease and the PSG data. For future studies it will be useful to correlate common OSAS risk factors and hearing loss and determine whether one or more of these factors correspond to a cause-and-effect relationship. In addition, even if the antioxidant therapy is demonstrated to have promising results [25,54], it is important to start with new clinical trials on antioxidant systemic effects in OSAS patients and observations on improved hearing function. We propose that for the early diagnosis of damage to hearing function, performing and evaluating extended high frequencies and/or performing vocal audiometry in a noisy environment associated with oxidative stress biomarkers may be useful [Figure 2]. In addition, we suggest that it would be interesting to compare over time the hearing function in patients treated with CPAP and that of those who refuse or cannot tolerate this therapy.

## Figures and Tables

**Figure 1 antioxidants-12-01430-f001:**
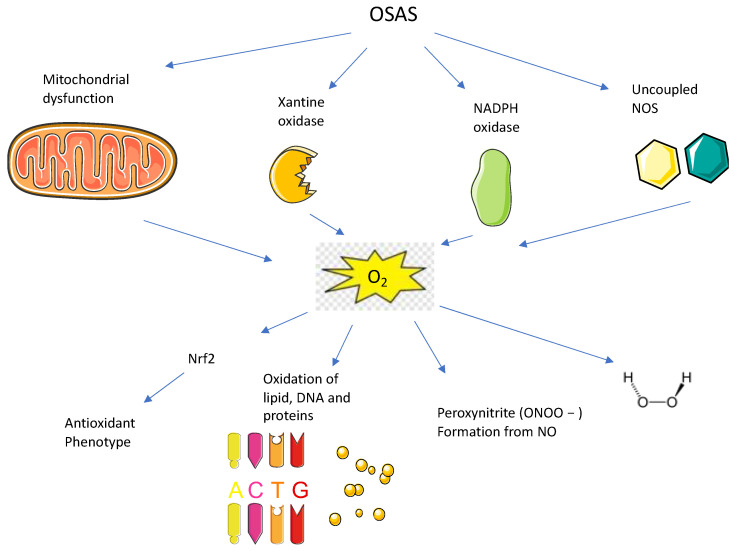
Pathways involved in OSAS endothelial damage due to oxidative stress and oxidative molecular changes (see text for description).

**Figure 2 antioxidants-12-01430-f002:**
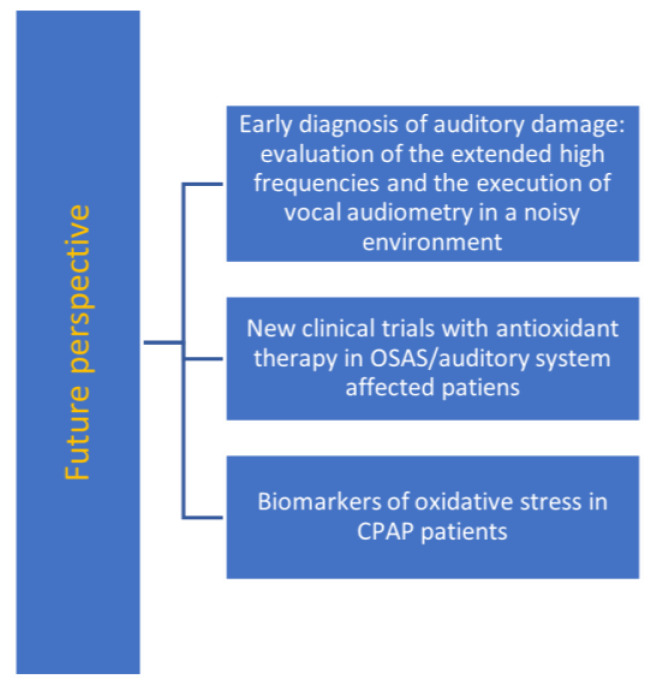
The integrated approach of translational medicine to OSAS disease and auditory system impairment (see text for description).

**Table 1 antioxidants-12-01430-t001:** Summary of OSAS articles, with investigations on the auditory system (see Results section for a discussion).

Study Title	Group/Patients	Age	Date
Qiuyang Fu, Tao Wang, Yong Liang et al.: Auditory Deficits in Patients with Mild and Moderate Obstructive Sleep Apnea Syndrome: A Speech Syllable Evoked Auditory Brainstem Response Study. Clin and Exp Otorhinol 12(1): 58–65 [26].	52 (31 OSAS + 21 ctrl)	23 years to 39 years	2019
Mustafa Sitki Gozeler, Furkan Sengoz: Auditory Function of Patients with Obstructive Sleep Apnea Syndrome: A Study. Eur-asian J Med 52(2): 176-9 [27].	65 (35 OSAS + 30 ctrl)	39 years to 48 years	2020
Spinosi MC, D’Amico F, Passali G, Cingi C, Rodriguez H, Passali D. Hearing loss in mild OSAS and simple snoring patients. Otolaryngol Pol. Apr 30;71(2):11–15 [28].	80 (50 OSAS + 30 ctrl)	45 years to 65 years	2017
Martines F, Ballacchino A, Sireci F, Mucia M, La Mattina E, Rizzo S, Salvago P. Audiologic profile of OSAS and simple snoring patients: the effect of chronic nocturnal intermittent hypoxia on auditory function. Eur Arch Otorhinolaryngol. Jun;273(6):1419-24 [29].	160 (100 OSAS + 60 ctrl)	38 years to 55 years	2016
Manuele Casale, Emanuela Vesperini, Massimiliano Potena et al.: Is obstructive sleep apnea syndrome a risk factor for auditory pathway? Sleep Breath 16:413–417 [30].	60 (39 OSAS + 21 ctrl)	31 years to 39 years	2012
Wei Wang, Jiao Su, Delei Kong et al.: Gender, nocturnal hypoxia, and arousal influence brainstem auditory evoked potentials in patients with obstructive sleep apnea. Sleep Breath [31].	118 (84 OSAS + 34 ctrl)	53 years to 60 years	2016
Ayşe Iriz, Mehmet Duzlu, Oğuz Kokturk et al.: The effect of obstructive sleep apnea syndrome on the central auditory system. Turk J Med Sci; 48: 5–9 [32].	31 (21 OSAS + 10 ctrl)	47 years to 55 years	2018
Deniz, Ersözlü T. Evaluation of the changes in the hearing system over the years among patients with OSAS using a CPAP device. Cranio. 40(6):524–527 [33].	22 (OSAS under CPAP treatment)	56 years to 67 years	2022

## Data Availability

Not applicable.

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
