# Peer review of "Oxidative Stress in Obstructive Sleep Apnea Syndrome: Putative Pathways to Hearing System Impairment"

_antioxidants, 2023, doi:10.3390/antiox12071430_

Round 1

Reviewer 1 Report

This review by Pierluigi Mastino and ten co-authors is entitled: "Oxidative Stress in Obstructive Sleep Apnea Syndrome: Putative Pathways to the hearing system impairment"

Obstructive Sleep Apnea has been the subject of numerous studies and research over the past few decades.

This review is an extensive collection of published results and lacks novelty. The conclusions drawn by the authors are largely based on well-established facts and research, and lack substantial new insights.

Materials and Methods described a severely limited approach: search keywords [OSAS, auditory, hearing, audiology, CPAP, and oxidative] yielded 11 articles, 8 articles were selected and 7 were appropriate. However, the reference list has 64 citations.

The key concept in the field of Antioxidants can be found in the Abstract: "oxidative stress is involved in the damage of the hearing system" (lines 23–24).

However, the main goal "of this review is to demonstrate a correlation between OSAS and the auditory system" (lines 118-119).

The Introduction provides a comprehensive overview of the obstructive sleep apnea clinical research.

Results describe OSA-related impairment in the hearing system.

Discussion says that OSA-related impairment of the auditory system is "the object of debate".

Antioxidant-related issues appear only in Discussion and in Figure 2.

In all, I did not find much of the deep elaboration or in-depth analysis of the existing literature. What are the most exciting new hypotheses or ideas that came out of this review article?

Figure 2 could be the center of the concept, and it has to be explained in more details.

Minor.

What message does Figure 1 convey?

Author Response

Dear Editor,

We thank the editor for his/her letter and the reviewers for their comments on our manuscript (Manuscript ID: diseases-2462652). Those comments are all valuable and very helpful for revising and improving our paper, as well as the important guiding significance to our research. We have studied the comments carefully and have made corrections which we hope meet with approval. Revised portions are marked in yellow on the paper. Please see below for point-by-point responses.  The main corrections in the manuscript and the response to the reviewer's comments are as follows:

Reply to the Reviewer # 1 comments:

This review by Pierluigi Mastino and ten co-authors is entitled: "Oxidative Stress in Obstructive Sleep Apnea Syndrome: Putative Pathways to the hearing system impairment"

Obstructive Sleep Apnea has been the subject of numerous studies and research over the past few decades.

This review is an extensive collection of published results and lacks novelty. The conclusions drawn by the authors are largely based on well-established facts and research, and lack substantial new insights.

Comment Materials and Methods described a severely limited approach: search keywords [OSAS, auditory, hearing, audiology, CPAP, and oxidative] yielded 11 articles, 8 articles were selected and 7 were appropriate. However, the reference list has 64 citations.

Response Materials and Methods section was rewritten, indicating many other selected and evaluated articles; the keywords searched and selection criteria adopted.

Line 131-149: A literature review has been conducted. Pubmed research with keywords such as “OSAS”, “auditory”, “hearing”, “audiology”, “CPAP”, and “oxidative” has been performed. “OSAS and auditory” retrieved 18 articles, whereas ‘OSAS and auditory and oxidative’ no article has been found; “OSAS and hearing” retrieved 57 articles, and with “oxidative” no results were retrieved; “OSAS and audiology” showed 42 results, but only one with “oxidative”. Lastly, “CPAP and oxidative” retrieved 206 articles, and 15 articles have been selected.  All the published articles with pediatric patients have been excluded because this study is based only on adult patients. Furthermore, all the research that included all the comorbidities OSAS related, like cardiovascular and metabolic diseases, or every condition that could affect the auditory system, haven’t been considered. This is because, to demonstrate the correlation between OSAS and hearing loss, it is necessary to exclude other causes of hearing damage pathology related. Similarly, all the articles with patients affected by ORL diseases, like otitis media, tubaric dysfunctions, and nasal obstruction, that can cause neurosensorial, conductive, or mixed hearing loss, have been excluded. In the end, all the studies with lab animals involved, have been eliminated from the study because our research is focused only on human subjects. From this literature review, only eleven articles were considered, but only eight of which evaluated extensively OSAS effects on the auditory system and how this can be linked to microvascular damage of the tissues, and one focused on the effects of CPAP treatment.

Comment The key concept in the field of Antioxidants can be found in the Abstract: "oxidative stress is involved in the damage of the hearing system" (lines 23–24).

Response Thanks again for the reviewer's comments. We reorganize the key concept by adding several sentences to the Discussion section as reported below.

Comment However, the main goal "of this review is to demonstrate a correlation between OSAS and the auditory system" (lines 118-119).

Response Thanks again for the reviewer's comments. We reorganize the key concept by adding several sentences to the Discussion section as reported below.

Comment The Introduction provides a comprehensive overview of the obstructive sleep apnea clinical research.

Response Thanks again for the reviewer's comments

Comment Results describe OSA-related impairment in the hearing system.

Response Thanks again for the reviewer's comments

Comment Discussion says that OSA-related impairment of the auditory system is "the object of debate".

Response Thanks again for the reviewer's comments. As reported above, the Discussion was entirely reorganized and several parts were added or rewritten from line 270-428 as follow:

“How OSAS can affect the auditory system? There are several hypotheses mainly associated with microvascular impairment and the damaging effects of reactive oxygen species (ROS). Steiner et al. [34] reported that blood plasma viscosity can be increased in OSAS patients, with an impairment of the microcirculation, and in fact, Bernard et al. evidenced that blood higher viscosity can result in a dysfunction of the hearing system [35]. Instead, it was suggested that exposure to continuous noise due to snoring can result in hearing loss [36]. The most accepted theory is that hypoxemia can be a negative factor in OSAS patients, compromising their hearing system. In OSAS disease, hypoxemia causes damage in multiple systems, like those cardiovascular and neurologic. From animal studies was reported that intermittent ischemia causes auditory damage due to mitochondrial damage in the hair cells of the internal ear [37]. The internal ear represents an anatomic region extremely susceptible to anoxic/hypoxic damage, due to the formation of ROS and increased oxidative stress, that activates an inflammatory and an immune response leading to both vascular and metabolic complications [38] (Fig.1). Multiple pieces of evidence suggest that altered mitochondrial function and oxidative stress both in the OSAS and cochlea could play a pivotal role [25, 39-43]. These observations were reported individually in OSAS or hearing loss, but investigations aimed to explore the mitochondrial function linking OSAS and hearing system impairment are lacking.

4.1 Mitochondrial Dysfunction and ROS

Mitochondria are the major intracellular source of reactive oxygen species (ROS). ROS include free radicals such as superoxide anion (O2•−) and hydroxyl radical (HO•), and non-radicals, such as hydrogen peroxide (H2O2) [44]. ROS imbalance between the production and the level of antioxidant defense in the cell context drives oxidative stress. Therefore, oxidative stress causes irreversible ROS-mediated cellular damage to DNA, proteins, and lipids [45]. The role of mitochondria in recurrent hypoxia is not evidenced only in neuronal cells, but also in genioglossus and palatine muscles [46]. Mitochondria are the main producers of ROS during reoxygenation [47] (Fig.2). OSAS patients present a reduction of blood mitochondrial DNA levels, and it’s a marker of mitochondrial damage [37]. Another source of ROS, during the reoxygenation phase, is the production of superoxide due to enzymes like xanthine oxidase, nitric oxide synthase, and NADPH oxidase [48].  ROS stimulates the expression of adhesion molecules, like L-selectin and integrins, and related endothelial adhesion molecules, like E-selectin, P-selectin, ICAM-1, and VECAM-1 that leads to microvascular damage [49,50]. An antioxidant system is a set of endogenous defense mechanisms with the aim to protect the organism against radical damage, characterized by enzymes like superoxide dismutase, catalase, peroxidase, and molecules like glutathione, vitamin C, and vitamin E [51]. The imbalance between increased ROS levels and ineffective antioxidant capacity can be quantified by multiple biomarkers, which are proteins generated by the oxidation of nucleic acids, proteins, and lipids [52-55].

4.2 Hypoxia and HIF-α

Our body tries to adjust itself in this condition of hypoxia, by producing numerous molecules useful for cells survival in condition of lack of oxygen, like Hypoxia Induced Factor 1-α (HIF-1α) and Vascular Endothelial Growth Factor (VEGF) [56]. HIF-1α is one of the most important factors in oxygen’s homeostasis, in fact, its levels, like those of NF-kB, are related to the severity of the disease (AHI and ODI) [57]. HIF-α is involved in OSAS redox signaling that leads to multiple systemic and cellular functional changes, like blood pressure, increased release of neurotransmitters, and alteration in sleep and cognitive functions [58].  Sies et al. showed that HIF-α is activated by a different pathway in intermittent hypoxia than in prolonged hypoxia [59]. Intermittent hypoxia seems to cause less stability of HIF-α, leading to the activation of NF-kB, probably due to oxidative stress [60]. NF-kB is probably very important in the pathogenesis of OSAS because it’s under complex regulation by multiple regulatory molecules and coordinates the inflammatory response, like the production of adhesion molecules, cytokines, and adipokines [61]. Sterol regulatory element-binding proteins (SREBPs) are a group of transcription factors involved in lipid homeostasis that are redox sensitive [62]. Multiple studies demonstrated that hyperlipidemia mediated by SREBPs is involved in lipid peroxidation and atherosclerosis induced by intermittent hypoxia [63]. Lipid peroxidation causes impairment in membranes, lipoproteins, and other lipid molecules. Lipid peroxidation causes the production of multiple secondary molecules like some aldehydes [64]. In this regard, malondialdehyde (MDA), is the main product of these aldehydes in the peroxidation mechanism and it is a marker of oxidative stress in some lung diseases [65]. Pau et al. showed that MDA levels were higher in OSAS patients compared to non-OSAS individuals suggesting MDA as a potential biomarker of OSAS [66]. VE-cadherin cleavage is related to endothelial impairment in OSAS patients, because of increased values of its soluble form in the patient’s blood, leading to increased endothelial permeability and its association with other mechanisms involved with oxidative stress (ROS production, HIF-α, VEGF and tyrosine kinase pathways [67] (Fig.2). The production of extracellular vesicles by red blood cells is another factor involved in endothelial impairment and related to decreased eNOS, decreased Endothelin-1 (ET-1), and phosphorylation by the PI3K/AKT pathway [68].

4.3 OSAS, miRNAs, and auditory system

It seems also that some epigenetic alterations are involved in intermittent hypoxia and OSAS, involving the small non-coding-RNA as microRNA (miRNAs) [25]. MiRNAs in this era of medicine are considered ideal biomarkers [69] and has been found that in OSAS patients there are decreased levels of miR-199-3p, 107, and 485-5p and increased levels of miR574-5p [70]. In a murine model, has been evidenced that miR-155 induces oxidation and intensifies the NLRP3 inflammasome pathway induced by intermittent hypoxia, inhibiting the FOXO3a gene and HK-2 cells [71]. Moreover, miR-155 seems to have a proapoptotic function in some disease where there are decreasing level of antiapoptotic molecules, like clusterin, that has increased levels in OSAS patients and correlates with miR-155 [72]. MiR-664a-3p is a potential biomarker of atherosclerosis in OSAS because it has decreased levels in OSAS patients, and it’s negatively related to AHI and carotid intima-media maximum thickness [73]. Other micro-RNA, like miR-630 in pediatric OSAS, and miR-30a, miR-34a-5p, and miR-193 in murine studies, take part in endothelial dysfunction in OSAS patients [74,75]. It is to be noted that a deep analysis of altered expression of miRNAs in OSAS and auditory system, could open a new hypothesis to explore the redox unbalancing shared in this connection.

4.4 OSAS and Interleukins

Intermittent hypoxia and oxidative stress activate an immune response by our organism, involving proinflammatory molecules, like TNF, CRP, and IL-6 and IL-8, with increased levels of NF-kB and TNF-α [76]. IL-8 and IL-17 are linked to OSAS severity [77]. CRP and TNF- α levels are lower after surgery, but still higher than in healthy patients, as shown by Olszewska et al. [78]. Moreover, several peripheral blood cells are involved. Monocytes overexpress Toll-Like Receptor (TLR) in OSAS patients, and the TLR-6 gene is upregulated through methylation of the DNA [79]. Huang et al. [80], showed that site number 1 cytosine-phosphate-guanine (CPG) is hypermethylated in patients affected by severe OSAS. Macrophages also play a key role in this inflammatory process due to oxidative stress, producing numerous molecules and reactive oxygen species, like substances reactive to thiobarbituric acid, 8-OHdG, and asymmetric dimethylarginine [81]. Oxidative stress contributes to sleep behavior in patients with OSAS and some authors suggest that the intake of antioxidants improves sleep quality [25,82].  However, most of the molecules are not tested in humans, and those tested in humans have not been studied in cohorts large enough to give indications of their use. Vitamin C and N-acetylcysteine showed interesting results in the reduction of oxidative stress in OSAS [54], and patients who received it in chronic evidenced an improvement in sleep parameters. Moreover, has been demonstrated that leptin can reduce free radicals, oxidative stress, and atherosclerosis in OSAS patients [83].

4.5 Oxidative Stress and auditory system

Regarding the hearing system, the external ciliate cells of the cochlea, at its basal turn, are vulnerable to oxidative stress, probably because there is a minor activity of the antioxidant enzymes correlated with glutathione [84]. It’s important to note that, according to the tonotopic theory, the basal turn encodes for the high frequencies. Excessive oxidative stress can lead to a dysfunction of the microcirculation, as showed by Patt et al. [85], which examined the endothelial function in OSAS patients not affected by cardiovascular diseases. They evidenced an increase in the peroxynitrite levels in the microvascular wall of the OSAS patients, leading to excessive production of NO and superoxide in the endothelial environment. This uptake of NO by the superoxide leads to a reduction in the availability of NO and then, to changes in the microcirculation that are independent of age, sex, and weight, and that are reversible with therapy.  Similar mechanisms can be responsible for the progressive damage of the internal ear, where the production of NO by the vascular cells of the cochlea leads to a relaxation of the smooth muscles and pericyte, inhibiting the voltage-gated calcium channels and activating the ATP-sensitive K+ channels in the endothelial and smooth muscles cell of the spiral modular artery [68]. Thus, cochlear damage and hearing loss can be early markers of impairment of the microcirculation in individuals affected by OSAS [86]. CPAP is the gold standard for OSAS treatment, and it’s used to overcome rhino- and oropharyngeal collapse in patients with moderate and severe OSAS. The purpose of the CPAP is to allow the patient, during sleep, to breathe and to increase oxygen levels. It’s important to know that CPAP is the only confirmed antioxidant therapy and has been shown that it’s capable of reversing some of the alterations induced by oxidative stress, like eNOS, nitro-tyrosine, and NF-kB in the endothelium and circulating TNF-α [87]. However, possible side effects can be aerophagy, sinusitis, dryness of the oral and nasal mucosa, congestion, sneezing, and epistaxis. In addition, unexpected and serious side effects have been reported, like pneumocephalus, pulmonary barotrauma, intraocular hypertension, subcutaneous emphysema, and hearing loss due to barotrauma. Quiyang et al. [26], in their study, reported that the transient component of the speech ABR is significantly and positively correlated to the AHI of patients with mild and moderate OSAS whereas, these changes, in the conventional click-ABR, cannot be evidenced. They suggest this test as a potential biomarker in the diagnosis and management of OSAS in the early stages. According to Gozeler and Sengoz [27], the hearing system is affected in various degrees in OSAS patients, detecting in general, mild neurosensorial hearing loss in OSAS patients with respect to healthy ones. Spinosi et al. [28], showed no correlation between hearing loss and exposure to chronic noise due to snoring, even in patients with mild OSAS. Martines et al. [29], underlined the key role of chronic intermittent hypoxia in the development of hearing dysfunctions and a more marked hearing loss in the high frequencies in patients with severe OSAS. Casale et al. [30], confirmed this theory, asserting that hypoxia could be a risk factor for the hearing system, observing a hearing dysfunction in their patients without other risk factors. Iriz et al. [32], similarly, showed that repeated hypoxic events can cause multisystemic disorders in OSAS patients, even in the hearing system. Wang et al. [31] evidenced some abnormalities in the brainstem auditory evoked potentials in patients with moderate and severe OSAS, obtaining longer latencies in the waves I, III, and V, especially in the right ear of male patients. In the end, Deniz and Ersozlu [33], have demonstrated that CPAP doesn’t improve the hearing function of OSAS patients, but it has no negative effects too, due to a possible barotrauma.”

Comment Antioxidant-related issues appear only in Discussion and in Figure 2.

Response We appreciate for the comment and the ‘Antioxidant-related issues’ appear also in the Introduction. Line 108-116: “The pathophysiology of these complications is not entirely elucidated but seems to involve multiple pathways, one of which is endothelial damage due to oxidative stress. This is defined as an imbalance between the pro-oxidant and antioxidant system, that leads to the excessive formation of reactive oxygen species (ROS). ROS represents the response of our organism to multiple insults, including oxidative stress, and in OSAS patients, is tightly related to hypoxia [23,24]. Oxidative stress is one of the most important features in the development of cardiovascular comorbidity. Oxidative stress and intermittent hypoxia could lead to multiorgan impairment and antioxidant therapy is promising, suggesting the need to discover new biomarkers for an early diagnostic framework of OSAS patients [25].”

Comment In all, I did not find much of the deep elaboration or in-depth analysis of the existing literature. What are the most exciting new hypotheses or ideas that came out of this review article?

Response We appreciated the reviewer's criticism, and new hypotheses or ideas are arguments in the Discussion section and in the Conclusion and Perspectives section. In addition, figure 3 was reorganized as a synopsis of future perspectives.

Comment Figure 2 could be the center of the concept, and it has to be explained in more details.

Response As above, the mechanism and molecules indicated in Fig. 2 were discussed in the revised manuscript.

Minor.

Comment What message does Figure 1 convey?

Response We eliminated Figure 1.

We appreciate for Editors/Reviewers’ warm work earnestly and really hope that our modification of this paper can get your precious recognition, which is of great significance to us.

Reviewer 2 Report

I appreciate the opportunity to review the manuscript for publication in MDPI Antioxidants.

I feel that the topics are interesting. How OSAS can affect the auditory system is still the object of debate. The manuscript is narrative but well organized.

I have a few comments as follows.

In Introduction, the authors had better more specifically state possible risk factors of the auditory function in OSAS patients as compared to non-OSAS patients.

In Results, most of the entire article describe and refrain their main outcomes as in the original literature. This gives the impression that the content is somewhat unfocused and tedious.

The cited articles should be summarized in a table.

L55: AHI should be typed out in the first appearance.

L129: A literature review has been conducted with keywords such as OSAS, auditory, hearing, audiology, CPAP, and oxidative. Please disclose the source of literature data base.

L325: “the external ciliate cells of the cochlea” should be the outer hair cells.

Author Response

Dear Editor,

We thank the editor for his/her letter and the reviewers for their comments on our manuscript (Manuscript ID: diseases-2462652). Those comments are all valuable and very helpful for revising and improving our paper, as well as the important guiding significance to our research. We have studied the comments carefully and have made corrections which we hope meet with approval. Revised portions are marked in yellow on the paper. Please see below for point-by-point responses.  The main corrections in the manuscript and the response to the reviewer's comments are as follows:

Reply to the Reviewer # 2 comments:

I appreciate the opportunity to review the manuscript for publication in MDPI Antioxidants.

I feel that the topics are interesting. How OSAS can affect the auditory system is still the object of debate. The manuscript is narrative but well organized.

I have a few comments as follows.

Comment In Introduction, the authors had better more specifically state possible risk factors of the auditory function in OSAS patients as compared to non-OSAS patients.

Response We added in the introduction section these sentences:

Line 120-128: ‘The hearing impairment risks factors, such as age, ototoxic drugs, noise, diabetes, altered lipid metabolism, smoking, and coronary heart disease, can be evaluated in OSAS patients with the aim to quantify more specifically state possible risk factors of the auditory function in OSAS patients as compared to non-OSAS patients. Several studies have focused on risk factors for OSAS or hearing loss, but no studies have been conducted for both combined and associated diseases. We suggest that OSAS itself could be a risk factor for hearing loss. Moreover, we discuss if the CPAP could improve patients hearing, or if its prolonged application, could worsen the auditory system. It will also explore the pathophysiology of auditory damage, showing several pathogenic theories.’

Comment In Results, most of the entire article describe and refrain their main outcomes as in the original literature. This gives the impression that the content is somewhat unfocused and tedious.

The cited articles should be summarized in a table.

Response We added a new Table 1.

Study Title

Group/patients

Age

Date

Qiuyang Fu, Tao Wang, Yong Liang et al: Auditory Deficits in Patients with Mild and Moderate Obstructive Sleep Apnea Syndrome: A Speech Syllable Evoked Auditory Brainstem Response Study. Clin and Exp Otorhinol 12(1): 58-65 [26].

52 (31 OSAS+21 ctrl)

23 years to 39 years

2019

Mustafa Sitki Gozeler, Furkan Sengoz: Auditory Function of Patients with Obstructive Sleep Apnea Syndrome: A Study. Eur-asian J Med 52(2): 176-9 [27].

65 (35 OSAS+30 ctrl)

39 years to 48 years

2020

Spinosi MC, D'Amico F, Passali G, Cingi C, Rodriguez H, Passali D. Hearing loss in mild OSAS and simple snoring patients. Otolaryngol Pol. Apr 30;71(2):11-15. [28].

80 (50 OSAS+30 ctrl)

45 years to 65 years

2017

Martines F, Ballacchino A, Sireci F, Mucia M, La Mattina E, Rizzo S, Salvago P. Audiologic profile of OSAS and simple snoring patients: the effect of chronic nocturnal intermittent hypoxia on auditory function. Eur Arch Otorhinolaryngol. Jun;273(6):1419-24. [29].

160 (100 OSAS+60 ctrl)

38 years to 55 years

2016

Manuele Casale, Emanuela Vesperini, Massimiliano Potena et al: Is obstructive sleep apnea syndrome a risk factor for audi-tory pathway?  Sleep Breath 16:413–417 [30].

60 (39 OSAS+21 ctrl)

31 years to 39 years

2012

Wei Wang, Jiao Su, Delei Kong et al: Gender, nocturnal hypoxia, and arousal influence brainstem auditory evoked potentials in patients with obstructive sleep apnea. Sleep Breath [31].

118 (84 OSAS+34 ctrl)

53 years to 60 years

2016

Ayşe Iriz, Mehmet Duzlu, Oğuz Kokturk et al: The effect of obstructive sleep apnea syndrome on the central auditory system. Turk J Med Sci; 48: 5-9 [32].

31 (21 OSAS+10 ctrl)

47 years to 55 years

2018

Deniz, Ersözlü T. Evaluation of the changes in the hearing system over the years among patients with OSAS using a CPAP device. Cranio. 40(6):524-527 [33].

22 (OSAS under CPAP treatment)

56 years to 67 years

2022

Table 1. Summary of OSAS articles, with investigations on the auditory system (see Results for discussion).

Comment L55: AHI should be typed out in the first appearance.

Response L50 apnoea hypopnoea index (AHI), was typed.

Comment L129: A literature review has been conducted with keywords such as OSAS, auditory, hearing, audiology, CPAP, and oxidative. Please disclose the source of literature data base.

Response Materials and Methods section was rewritten, indicating many other selected and evaluated articles; the keywords searched and selection criteria adopted.

Line 131-149: A literature review has been conducted. Pubmed research with keywords such as “OSAS”, “auditory”, “hearing”, “audiology”, “CPAP”, and “oxidative” has been performed. “OSAS and auditory” retrieved 18 articles, whereas ‘OSAS and auditory and oxidative’ no article has been found; “OSAS and hearing” retrieved 57 articles, and with “oxidative” no results were retrieved; “OSAS and audiology” showed 42 results, but only one with “oxidative”. Lastly, “CPAP and oxidative” retrieved 206 articles, and 15 articles have been selected.  All the published articles with pediatric patients have been excluded because this study is based only on adult patients. Furthermore, all the research that included all the comorbidities OSAS related, like cardiovascular and metabolic diseases, or every condition that could affect the auditory system, haven’t been considered. This is because, to demonstrate the correlation between OSAS and hearing loss, it is necessary to exclude other causes of hearing damage pathology related. Similarly, all the articles with patients affected by ORL diseases, like otitis media, tubaric dysfunctions, and nasal obstruction, that can cause neurosensorial, conductive, or mixed hearing loss, have been excluded. In the end, all the studies with lab animals involved, have been eliminated from the study because our research is focused only on human subjects. From this literature review, only eleven articles were considered, but only eight of which evaluated extensively OSAS effects on the auditory system and how this can be linked to microvascular damage of the tissues, and one focused on the effects of CPAP treatment.

Comment L325: “the external ciliate cells of the cochlea” should be the outer hair cells

Response L285 the terms ‘outer hair cells’ were typed.

Round 2

Reviewer 1 Report

The authors refined their paper and addressed some of the issues I pointed out in my review.

In my previous review, I noted: "The key concept in the field of Antioxidants can be found in the Abstract: "oxidative stress is involved in the damage of the hearing system" (lines 23–24). However, the main goal "of this review is to demonstrate a correlation between OSAS and the auditory system" (lines 118-119)." The key concept (oxidative stress) has to be linked to the main goal (OSAS and the auditory system). Please clarify the relationship between Obstructive Sleep Apnea Syndrome and oxidative stress in the auditory system.

I noted: "The Introduction provides a comprehensive overview of the obstructive sleep apnea clinical research." What is the relevance of this to the primary objective? I suggest to introduce the antioxidant-related issues (already in Introduction), which appeared only in Discussion. 

Response We reorganize the key concept by adding several sentences to the Discussion section as reported below.

The key concept can be found in the Abstract: "oxidative stress is involved in the damage of the hearing system", and it has to be included in the Introduction section.

Response : ‘Antioxidant-related issues’ appear also in the Introduction. Line 108-116... 

The authors said that oxidative stress is involved in many pathologies, such as "cardiovascular comorbidity" etc. Indeed, oxidative stress has been implicated in the development of a number of diseases and conditions, including heart disease, cancer, diabetes, and aging.  Could the authors be more precise and concentrate on the oxidative mechanisms "involved in the damage of the hearing system in OSA"?

In my previous review, I noted: "Results describe OSA-related impairment in the hearing system". In Results, the clinical data did not indicate a correlation between oxidative stress and the patients' symptoms. The summary of reports in Table 1 regarding the auditory system in OSAS indicates that the current paper is comprised of existing published results and lacks novelty. Table 1 displays the titles of articles and the number of participants, but does not provide insight as to the involvement of the auditory system in OSAS.

I asked: "What are the most exciting new hypotheses or ideas that came out of this review article?"

Response : new hypotheses or ideas are arguments in the Discussion section and in the Conclusion and Perspectives section. In addition, figure 3 was reorganized as a synopsis of future perspectives.

I did not receive an answer about novelty (i.e. I missed new exciting hypotheses or ideas). Figure 3 does not exist. I found Figure 2 with a diagram entitled "Future perspectives of an integrated approach of translational medicine".

I asked: "What message does Figure 1 convey?"

Figure 1 could be the center of the concept, and it has to be explained in more details. The figure legend refers to a text explaining the figure's contents, but this text does not exist. The caption of Figure 1 should provide a concise description of what is depicted in the figure.

I'm sorry to say it again: I did not find much of the deep elaboration or in-depth analysis of the existing literature.

Author Response

Dear Editor,

We thank the reviewer for their comments on our manuscript (Manuscript ID: diseases-2462652). This second round of reply comments is aimed to improve the manuscript. Revised portions are marked in yellow and green on the paper. We hope that our effort to explain some novelties and a "profound elaboration or in-depth analysis of the existing literature" as required, will be satisfactory. The main corrections in the manuscript and the response to the reviewer's comments are as follows:

ROUND 1/2: Reviewer comments

The authors refined their paper and addressed some of the issues I pointed out in my review.

In my previous review, I noted: "The key concept in the field of Antioxidants can be found in the Abstract: "oxidative stress is involved in the damage of the hearing system" (lines 23–24). However, the main goal "of this review is to demonstrate a correlation between OSAS and the auditory system" (lines 118-119)." The key concept (oxidative stress) has to be linked to the main goal (OSAS and the auditory system). Please clarify the relationship between Obstructive Sleep Apnea Syndrome and oxidative stress in the auditory system.

I noted: "The Introduction provides a comprehensive overview of the obstructive sleep apnea clinical research." What is the relevance of this to the primary objective? I suggest to introduce the antioxidant-related issues (already in Introduction), which appeared only in Discussion. 

Response We reorganize the key concept by adding several sentences to the Discussion section as reported below.

Response 2: In the introduction were rephrased and reorganized following sentences, lines 116-136:

“The cochlear hair cells are most vulnerable to oxidative stress, particularly those located at the base of the cochlea itself, leading to sensorineural hearing loss (SNHL), especially for high frequencies, in response to multiple causes, such as ototoxic agents, noise exposure and aging. Antioxidant therapy is promising, suggesting the need to discover new biomarkers for an early diagnostic framework of OSAS patients [25]. Antioxidant therapy has proven to be effective also for acquired disorders that induce SNHL [26]. The objective of this review is to explore a correlation between OSAS and the auditory system. More specifically our aim is to underline that an untreated OSAS could negatively affect the auditory function as compared to non-OSAS patients. The hearing impairment risks factors, such as age, ototoxic drugs, noise, diabetes, altered lipid metabolism, smoking, and coronary heart disease, can be evaluated in OSAS patients with the aim to quantify more specifically state possible risk factors of the auditory function in OSAS patients as compared to non-OSAS patients. Several studies have focused on risk factors for OSAS or hearing loss, but no studies have been conducted for both combined and associated diseases. We suggest that OSAS itself could be a risk factor for hearing loss. In the same way, antioxidant therapy seems to be effective in OSAS and in SNHL, but there isn’t evidence of the effectiveness of antioxidant therapy in OSAS patients with SNHL, although it could be probably successful.  Moreover, we discuss if the CPAP could improve patients hearing, or if its prolonged application, could worsen the auditory system. It will also explore the pathophysiology of auditory damage, showing several pathogenic theories.”

The key concept can be found in the Abstract: "oxidative stress is involved in the damage of the hearing system", and it has to be included in the Introduction section.

Response: ‘Antioxidant-related issues’ appear also in the Introduction. Line 108-116... 

Response 2R: This concept is included in the new version of the introduction.

The authors said that oxidative stress is involved in many pathologies, such as "cardiovascular comorbidity" etc. Indeed, oxidative stress has been implicated in the development of a number of diseases and conditions, including heart disease, cancer, diabetes, and aging.  Could the authors be more precise and concentrate on the oxidative mechanisms "involved in the damage of the hearing system in OSA"?

Response 2R: We refer to selected studies reported in the text and References. For example, the “cardiovascular comorbidity" was described in ref. 23 and 35.

In my previous review, I noted: "Results describe OSA-related impairment in the hearing system". In Results, the clinical data did not indicate a correlation between oxidative stress and the patients' symptoms. The summary of reports in Table 1 regarding the auditory system in OSAS indicates that the current paper is comprised of existing published results and lacks novelty. Table 1 displays the titles of articles and the number of participants, but does not provide insight as to the involvement of the auditory system in OSAS.

Response 2R: As requested in round 1 we inserted a new table, Table 1, where was reported the list of the studies discussed in the manuscript. To avoid duplicating this information in the table and manuscript text, the observations about the involvement of the auditory system in OSAS are only in the text.

I asked: "What are the most exciting new hypotheses or ideas that came out of this review article?"

Response: new hypotheses or ideas are arguments in the Discussion section and in the Conclusion and Perspectives section. In addition, figure 3 was reorganized as a synopsis of future perspectives.

Response 2R:  We improved section 4.1 Mitochondrial Dysfunction and ROS, adding several sentences, line from 309-331:

Mitochondria are intracellular organelles that have the function of nutrient metabolization, and ATP production and are involved in energy metabolism, generation of free radicals, calcium homeostasis, cell survival, and death [45]. They produce most of the energy of the body in the form of ATP through the Tricarboxylic acid cycle (TCA cycle) and the electron transport chain (ETC). Mitochondria are the major intracellular source of reactive oxygen species (ROS). The electron’s flow through the ETC is an imperfect process, that leads to an incomplete reduction of the oxygen by mitochondria and to the production of ROS [46]. ROS include free radicals such as superoxide anion (O2•−) and hydroxyl radical (HO•), and non-radicals, such as hydrogen peroxide (H2O2) [47]. Has been shown that the interaction between hydroxyl radicals and DNA causes damage on multiple sites of the DNA itself [48]. ROS imbalance between the production and the level of antioxidant defense in the cell context drives oxidative stress. This oxidative stress causes damage to the mitochondria, leading to an interruption of its functions, such as the production of ATP [49]. There are several defense mechanisms with the aim to protect from oxidative stress which are included enzymatic molecules such as superoxide dismutase, catalase, glutathione reductase, glutathione peroxidase, and non-enzymatic molecules, such as vitamin E and C, glutathione, carotenoids, and flavonoids.  Normally, the production of ROS remains in the mitochondria to protect the cells from oxidative damage. However, when the production of ROS overcomes the antioxidant defenses, oxidative stress causes irreversible ROS-mediated cellular damage to DNA, proteins, and lipids [50]. This mechanism damages the respiratory chain and causes mitochondrial dysfunction and leads to multiple pathological conditions like aging, metabolic disorders, and neurodegenerative pathologies.”

I did not receive an answer about novelty (i.e. I missed new exciting hypotheses or ideas). Figure 3 does not exist. I found Figure 2 with a diagram entitled "Future perspectives of an integrated approach of translational medicine".

Response 2R: Line 534, Figure 2 entitled: “The integrated approach of translational medicine to OSAS disease and auditory system impairment. (see text for description).”

I asked: "What message does Figure 1 convey?"

Figure 1 could be the center of the concept, and it has to be explained in more details. The figure legend refers to a text explaining the figure's contents, but this text does not exist. The caption of Figure 1 should provide a concise description of what is depicted in the figure.

I'm sorry to say it again: I did not find much of the deep elaboration or in-depth analysis of the existing literature.

Response 2R: We are sorry that the reviewer does not find novelty or deep elaboration or in-depth analysis of the existing literature. To improve the manuscript section 5. Conclusions and perspectives were entirely rewritten and reorganized from lines 456-531 and the references were added and discussed (from 87 to 100).

5.1 High-frequency pure tone audiometry

In our opinion, high-frequency pure tone audiometry could be a more cost-effective tool for an early diagnosis of hearing loss in OSAS patients. According to the tonopicity theory and to the pathophysiology of the possible auditory damage due to oxidative stress caused by intermittent hypoxia in OSAS patients, extended high frequencies might be impaired first by endothelial damage. High-frequency pure tone audiometry is a tool used to evaluate hearing threshold at the frequency range of 8-20 kHz and can detect hearing loss before the involvement of medium and low frequencies and so the impairment of hearing capacity (Fig.2).

5.2 The Role of antioxidant therapy

Antioxidants are molecules that inhibit ROS production and regulate oxidative stress. They can be endogenous, produced in vivo, and exogenous, taken from the outside. Endogenous antioxidants are the molecules we discussed early in this section. Exogenous antioxidants comprehend water and lipid-soluble molecules. Water-soluble antioxidants are methionine, vitamin C, carnitine, riboflavin, niacin, folic acid, polyphenols, and catechins. Methionine has the function to reduce cholesterol blood levels and to remove ROS [92]. Riboflavin and niacin eliminate lipid peroxides, with the aid of the GSH [93]. Vitamin C reduces toxicity by removing hydroxyl radicals [94]. Folic acid lowers homocysteine levels [95]. Lipid soluble antioxidants are β-carotene, vitamin E, astaxanthin, and coenzyme q10. Vitamin E gives stability to the biological membranes. Coenzyme q10 lowers the levels of vitamin E radicals after ROS removal [96]. There are some antioxidants that are both water and fat-soluble, like Gingko Biloba and alpha lipoic acid. The latter has both antioxidant properties and restores the antioxidant ability of glutathione, vitamin A; vitamin E, and vitamin C [97]. Has been demonstrated that vitamins C and E are potential treatments of SNHL. In fact, seems that vitamins improve hearing function in patients with sudden hearing loss [98]. Has been found, in an animal study, that vitamin E protects hair cells from the ototoxic damage of the cisplatin [99]. For what concerns OSAS patients, has been shown that vitamin C and N-acetylcysteine reduce oxidative stress [58]. N-acetylcysteine reduces oxidative stress in OSAS patients by reducing peroxidized lipids and increasing glutathione. Furthermore, an improvement in PSG data has been found [86]. Vitamin C improves endothelial function in OSAS patients [100]. Lastly, has been found that leptin, in OSAS patients, reduces free radicals, oxidative stress, and atherosclerosis [87]. It’s clear that antioxidant therapy could be effective in OSAS and in SNHL due to problems like aging, ototoxic drugs, and noise exposure. However, in our review, no evidence of the effectiveness of antioxidant therapy in patients affected by OSAS and SNHL combined has been found (Fig.2).

5.3 Perspectives

……….. In this study, it has been underlined that OSAS negatively affects the auditory system and there is a correlation with the severity of the disease and the PSG data. It would be interesting for future studies to correlate risk factors for OSAS and hearing loss and if one or more of them are corresponding in a cause-effect relationship. In addition, even if the antioxidant therapy is demonstrated to have promising results [25,54], it’s important to start with new clinical trials on antioxidant systemic effects in OSAS patients and observations on improved hearing function. We propose that for an early diagnosis of hearing function damage, could be useful the execution, in future studies, the evaluation of the extended high frequencies and/or the execution of vocal audiometry in a noisy environment and looking for potential biomarkers of oxidative stress for early detection of auditory damage [Fig. 2]. In addition, we suggest that it would be interesting to compare over time the hearing function in patients treated with CPAP and those who refuse or don’t tolerate this therapy.

We appreciate for Editors/Reviewers’ warm work earnestly and really hope that our modification of this paper can get your precious recognition, which is of great significance to us.